# How does individualised physiotherapy work for people with low back pain? A Bayesian Network analysis using randomised controlled trial data

**Bernard X. W. Liew** [1]*, **Jon J. Ford**[2], **Marco Scutari**[3], **Andrew J. Hahne**[2]

**1** School of Sport, Rehabilitation and Exercise Sciences, University of Essex, Colchester, Essex, United Kingdom, **2** Discipline of Physiotherapy, School of Allied Health, Human Services & Sport, La Trobe University, Melbourne, Australia, **3** Istituto Dalle Molle di Studi sull'Intelligenza Artificiale (IDSIA), Lugano, Switzerland

* bl19622@essex.ac.uk, liew_xwb@hotmail.com

**Data Availability Statement:** Data Available from: https://bernard-liew.github.io/2020_LBPcausal/index.html. Additionally, participants from the

## Abstract

### Purpose

Individualised physiotherapy is an effective treatment for low back pain. We sought to determine how this treatment works by using randomised controlled trial data to develop a Bayesian Network model.

### Methods

300 randomised controlled trial participants (153 male, 147 female, mean age 44.1) with low back pain (of duration 6–26 weeks) received either individualised physiotherapy or advice. Variables with potential to explain how individualised physiotherapy works were included in a multivariate Bayesian Network model. Modelling incorporated the intervention period (0–10 weeks after study commencement–"early" changes) and the follow-up period (10–52 weeks after study commencement–"late" changes). Sequences of variables in the Bayesian Network showed the most common direct and indirect recovery pathways followed by participants with low back pain receiving individualised physiotherapy versus advice.

### Results

Individualised physiotherapy directly reduced early disability in people with low back pain. Individualised physiotherapy exerted indirect effects on pain intensity, recovery expectations, sleep, fear, anxiety, and depression *via* its ability to facilitate early improvement in disability. Early improvement in disability, led to an early reduction in depression both directly and via more complex pathways involving fear, recovery expectations, anxiety, and pain intensity. Individualised physiotherapy had its greatest influence on early change variables (during the intervention period).

study provided informed consent that their data would only be made available to other researchers on specific requests to Dr. Ford or Dr. Hahne. We are therefore unable to publicly publish the dataset from this study, but reasonable requests to access the data can be directed either to the authors or alternatively to: Senior Manager, Ethics, Integrity and Biosafety Human Ethics Committee La Trobe University Telephone: +61 3 9479 1443 Email: humanethics@latrobe.edu.au Quote ethics ID: FHEC08/196.

**Funding:** LifeCare Health provided facilities, personnel and resources to allow treatment of trial participants free of charge in the STOPS Trial.

**Competing interests:** The authors have declared that no competing interests exist.

## Conclusion

Individualised physiotherapy for low back pain appears to work predominately by facilitating an early reduction in disability, which in turn leads to improvements in other biopsychosocial outcomes. The current study cannot rule out that unmeasured mechanisms (such as tissue healing or reduced inflammation) may mediate the relationship between individualised physiotherapy treatment and improvement in disability. Further data-driven analyses involving a broad range of plausible biopsychosocial variables are recommended to fully understand how treatments work for people with low back pain.

## Trials registration

ACTRN12609000834257.

## Introduction

Understanding how treatments work for people with low back pain (LBP) allows therapists to target factors most likely to facilitate recovery [1]. It is hypothesised that persistent low back pain is driven by a range of complex mechanisms that are not fully understood [2, 3]. Research has highlighted the potential importance of biological (eg. pathoanatomical [4], sleep [5]), psychological (eg. anxiety [6], catastrophizing [7]), and social factors such as work engagement [8]. Treatments that target a single mechanism of effect have typically demonstrated small and temporary treatment effects at best for people with LBP [9, 10]. Treatments such as stratified or individualised physiotherapy provide the opportunity to target multiple mechanisms of effect to optimise treatment outcomes [11, 12]. Randomised controlled trials (RCTs) of stratified [13] and individualised [14–16] physiotherapy have demonstrated small but promising effects in the treatment of people with LBP. Understanding the potential mechanisms driving short and long-term treatment effects provides an opportunity to better target treatment for individuals with LBP.

Previous research has used linear regression or structural equation modelling to evaluate how individualised [17] or stratified [18] physiotherapy might work for people with low back pain. These models examined the influence of hypothesised mediators (pain intensity, catastrophising, fear, anxiety, depression, stress, self-efficacy, coping and sleep) on the outcomes of pain and disability [17, 18]. A limitation of these previous mediation analyses is that structural assumptions must be made around the direction of mediation (eg. that psychosocial factors mediate disability but not vice-versa [17, 18]). In addition, previous mediation analyses have not allowed for mediators to influence each other [17, 18], like the sequential path of the fear-avoidance model (FAM) [19]. The recovery pathways underpinning the effects of individualised treatment for LBP could feasibly involve multiple interacting relationships between variables and the direction of those relationships may not be definitively known.

Bayesian Networks (BN) [20] are a class of probabilistic network modelling that provide a data-driven approach to derive complex pathways of effects. Unlike hypothesis-driven methods such as structural equations modelling (SEM) (e.g. in [21]), and simple mediation analysis (e.g. in [17]) that enforce a fixed structure, BN do not need to impose structural assumptions on the data informed by prior knowledge. BN may therefore reveal new relationships that have not been previously hypothesised. BN have previously been used to determine recovery

pathways for some musculoskeletal disorders including whiplash [22] and postoperative cervical radiculopathy [23], but not for individuals with LBP.

To explore how treatments work for people with LBP using a BN approach, data from a large randomised controlled trial is required. The Specific Treatment of Problems of the Spine (STOPS) Trial was a large RCT (n = 300) which showed that individualised physiotherapy was more effective than advice for people with LBP [16]. As a treatment informed by the biopsychosocial approach, several plausible recovery pathways could have underpinned the benefits of individualised physiotherapy. The aim of this study was to determine *how* the individualised physiotherapy (relative to the advice) approach in the STOPS RCT helped people with LBP.

## Materials and methods

### Study design

This Bayesian Network analysis was undertaken using data from the STOPS trial, which was a multicentre, parallel-group, randomised controlled trial completed in Melbourne, Australia. The STOPS trial enrolled 300 participants with LBP and has been reported previously [16, 24]. The trial was registered ACTRN12609000834257. The La Trobe University Human Ethics Committee approved the trial protocol [24]. All participants provided written informed consent to participate in the trial.

### Participants

To be included in the trial, participants needed to have a current episode of low back pain (and/or referred leg pain) between 6 weeks and 6 months duration, be aged 18–65 years and speak English [16]. Exclusion criteria were: a compensation claim, serious pathology (active cancer, cauda equine syndrome, foot drop making walking unsafe), pregnancy or childbirth within the last 6 months, history of lumbar spine surgery, spinal injections in the past six weeks, pain intensity < 2/10 or minimal activity limitation.

### Interventions (10 weeks)

Participants were randomised (via an offsite randomisation service) to receive either individualised physiotherapy (n = 156) or guideline-based advice (n = 144). Treatment was administered over a 10-week period. Participants were then discharged to independent self-management, but had outcomes followed up at 52 weeks.

**Guideline based advice.**   Participants allocated to guideline-based advice received 2 x 30-minute sessions over a 10-week period based on the approach described previously [25]. This included an explanation of the hypothesized source of the participant's pain, reassurance regarding a favourable prognosis, advice to remain active, and instruction regarding appropriate lifting techniques [24].

**Individualised physiotherapy.**   Participants allocated to the individualised physiotherapy group received 10 x 30-minute physiotherapy sessions over a 10-week period. Physiotherapy was individualised based on the patient's presentation. Available treatment components included pathoanatomical or neurophysiological information, education, self-management strategies (posture, pacing, pain management, sleep management, relaxation strategies), inflammatory management strategies (anti-inflammatory medication, taping, avoidance of provocative movements/postures), exercise (specific muscle activation, goal-oriented graded activity/exercise), manual therapy, directional preference management, and cognitive-behavioral strategies. Full details of the treatment protocols have been published previously [24, 26–29].

## Variables included in the Bayesian Network

The treatment group (individual physiotherapy or advice) was included in the BN as this was the variable upon which participants were randomised. Other variables were chosen for inclusion in the BN if they were measured at baseline and at 10 and 52-week follow-up (allowing change scores to be computed), and could feasibly act as predictors, mediators, or outcomes based on previous literature (see Introduction). Variables 5–11 below represented values from the individual items of the Örebro Musculoskeletal Pain Questionnaire (Örebro) [30]. Using single-item questions to quantify psychological features has been applied in previous studies on LBP for clinical subgrouping [31], and prognosis [32], and there is evidence of strong concurrent validity for single-item questions against full questionnaires in this population [33]. In addition, the single-item measurement strategy has been used in similar network studies in individuals with chronic pain [34], and in the broader field of psychology [35]. The following variables were used to form a BN:

1. Group: the randomised allocation into the two treatment arms (individualised physiotherapy versus advice).

2. Disability: Oswestry Disability Index [36]. The score ranges from 0 (no disability) to 100 (maximal disability).

3. LBP intensity: Measured on the 0–10 numerical rating scale (NRS). The score ranges from 0 (no pain) to 10 (worst pain possible).

4. Leg pain intensity: Measured on the 0–10 numerical rating scale (NRS). The score ranges from 0 (no pain) to 10 (worst pain possible).

5. Pain coping: Question of the Örebro [30], *"Based on all the things you do to cope, or deal with your pain, on an average day, how much are you able to decrease it"*? The score ranges from 0 (can decrease completely) to 10 (cannot decrease at all).

6. Pain persistence (recovery expectations): Question of the Örebro [30], *"In your view, how large is the risk that your current pain may become persistent?"* The score ranges from 0 (no risk) to 10 (very large risk).

7. Depressive symptoms: Question of the Örebro [30], *"How much have you been bothered by feeling depressed in the past week?"* The score ranges from 0 (not at all) to 10 (extremely).

8. Anxiety symptoms: Question of the Örebro [30], *"How tense or anxious have you felt in the past week?"* The score ranges from 0 (not at all) to 10 (extremely).

9. Work expectations: Question of the Örebro [30], *"In your estimation, what are the chances that you will be working in 6 months?".* The score ranges from 0 (very large chance) to 10 (no chance).

10. Fear-avoidance beliefs: A summed total of the three fear-avoidance beliefs questions of the Örebro [30]. The total score ranges from 0 (no fear) to 30 (maximum fear)– 10 points per question.

11. Sleep: Question 24 of the Örebro [30]. *"I can sleep at night"* Question scoring was inverted so that score ranges from 0 (can do it without pain being a problem) to 10 (cannot do it because of pain).

To quantify changes over time in variables 2 to 11, we coded time in the following manner:

1. Early changes (during the intervention period): The values at week 10 were subtracted from the values at baseline (week 10 minus baseline), and the resultant variables were suffixed with "_early".

2. Late changes (following discharge from treatment): The values at week 52 were subtracted from the values at week 10 (week 52 minus week 10), and the resultant variables were suffixed with "_late".

## Statistical analysis

**Bayesian network analysis.** All analyses were performed in R software using the "bnlearn" package [37], with codes and results included in a public online repository [38]. BN is a graphical modelling technique [20] used increasingly in the health sciences to understand causal relationships. BN can handle some missing data [39], which makes them practical in longitudinal studies where data sets are often incomplete. Analogous to multivariate regression modelling, BN quantifies the relationships among a set of variables, but BN also uses a directed acyclic graph (DAG) to visually depict the model. In BN modelling, each variable (or "node") can be statistically associated with one or more other nodes, with conditional dependencies between nodes termed "arcs". Building a BN model using a data-driven approach involves two stages: 1) structure learning—identifying relevant arcs; and 2) parameter learning—estimating the parameters that regulate the strength and direction of relationships between nodes.

BN can include prior knowledge, sourced from the literature and experts, during the model building process. To avoid imposing unproven assumptions on the model, we only built-in constraints that were undebatable (e.g. late change scores cannot influence early change scores, and early change scores cannot influence treatment group allocation). We allowed variables within the same time period to influence each other, consistent with previous studies [21, 40].

We made use of model averaging to reduce the potential of including spurious relationships in the BN, using bootstrap resampling (*B = 200*) and performing structure learning on each of the resulting samples using Structural Expectation-Maximization (EM) [39]. Structural EM is a technique that can build BN models in the presence of missing data [39]. It does so by building an initial empty BN model using the original complete data, using it to impute missing data, and then repeatedly rebuilding the BN model using the imputed data sets until convergence occurs. We computed an "average" consensus DAG by selecting arcs that were present in > 50% of the bootstrapped samples, to create a sparse and interpretable network [41].

To determine the validity (how well the model fitted the data) of the trained model, validation was performed using nested 10-fold cross-validation (CV). This splits the training set into 10 approximately equal folds, trains the model on 9 folds using bootstrap resampling (as described above), and evaluates the model's performance on the 10th fold. A nested 10-fold CV reduces the likelihood of overestimating the validity of the model, given that the same data used for training the model was also not used for validating it. Model validity was defined by computing the Pearson correlation coefficient between the predicted and observed values of each continuous variable, which is the predictive equivalent of the traditional $R^2$ fit index used in multiple regression. The strength of correlation was categorized as negligible ($|r| \leq 0.30$), low ($|r| = 0.31$ to $0.50$), moderate ($|r| = 0.51$ to $0.70$), high ($|r| = 0.71$ to $0.90$) and very high ($|r| = 0.91$ to $1$) [42]. The greater the model's predictive performance, the greater the correlation between the predicted and the observed values of the modelled variables. We also calculated the mean absolute error (MAE) (average of the absolute difference between the actual and predicted values), mean squared error (MSE) (average of the squared difference between the

original and predicted values), and root-mean-squared error (RMSE) (square root of MSE), between the observed and predicted values.

**Probing the network.** Similar to other statistical models, BNs can be used to answer questions about the nature of the data that go beyond mere descriptive observations. In prognostic modelling, prediction of an outcome and its uncertainty is desired based upon knowing the values of the predictors. Similar to prognostic modelling, BNs can be used to elicit the range of plausible values a variable can take upon knowing the values of the surrounding variables–and this is called conditional probability query. For each conditional probability query, we sampled $10^4$ data points of the variables of interest to obtain precise probability estimates, using a technique known as "logic sampling" [20].

**Missing data management.** Eleven participants (3.6%) had missing data in > 50% of the 21 variables, and were excluded from analysis, leaving 289 participants for network modelling. Imputation methods for multivariate models may become less accurate with large numbers of missing variables, hence we excluded this small number of participants who had large amounts of missing data [43]. The proportion of missing data per variable in these remaining participants are included in the (S1 Fig).

## Results

The mean (SD) values for the baseline demographics, pain, and psychological characteristics are found in Table 1. All continuous change variables used for BN analysis are reported in Fig 1, whilst the raw mean values of these variables at the three time points (baseline, 10 and 52 weeks) can be found in the (S2 Fig).

Fig 2 shows the averaged BN consensus model learnt from 200 networks constructed from the data, with arcs depicted if they appeared in > 50% of the networks. A BN graph can be qualitatively interpreted like the graph of a SEM (e.g. [21]) and a DAG of a typical three-variable mediation model (e.g. [44]). The predictive correlations for all variables are included in

**Table 1. Baseline descriptive characteristics of cohort included in Bayesian Network analysis.**

| Baseline variable | Advice (n = 137) | Individualized PT (n = 152) | Missing-excluded (n = 11) | P.value |
|---|---|---|---|---|
| Age(years) | 44.69(12.33) | 42.67(12.02) | 44.19(11.75) | 0.368 |
| Sex-Female* | 67 | 74 | 6 | 0.931 |
| Sex-Male* | 70 | 78 | 5 | 0.931 |
| Weight(kg) | 77.41(15.79) | 77.26(17.77) | 174.38(6.07) | 0.195 |
| Height(cm) | 169.97(11.52) | 172.24(10.47) | 74.7(8.93) | 0.885 |
| Disability | 29.71(13.25) | 29.31(11.85) | 26.82(5.78) | 0.750 |
| Low back pain intensity | 5.48(1.94) | 5.29(1.97) | 4.77(1.47) | 0.415 |
| Leg pain intensity | 4.76(2.62) | 4.58(2.74) | 4.8(2.49) | 0.860 |
| Pain coping | 5.06(2.15) | 5.16(2.37) | 4.36(2.38) | 0.526 |
| Anxiety | 5.08(2.5) | 4.62(2.58) | 4(3.52) | 0.183 |
| Depression | 3.54(2.98) | 3.01(2.98) | 3.64(3.5) | 0.312 |
| Pain persistence | 7.26(2.17) | 6.74(2.36) | 6.55(3.17) | 0.128 |
| Work expectations | 2.01(3.24) | 1.63(2.95) | 5.36(4.18) | 0.001 |
| Sleep | 3.69(2.54) | 3.57(2.49) | 3.45(2.5) | 0.893 |
| Fear | 18.11(6.01) | 18.58(6.17) | 22.6(6.13) | 0.081 |

Abbreviations: PT–physiotherapy.

*—variables represent counts of participants. All other values reflect mean (one standard deviation).

P values for continuous variables derived from a one-way Analysis of Variance test, count variable from a Chi-squared Test.

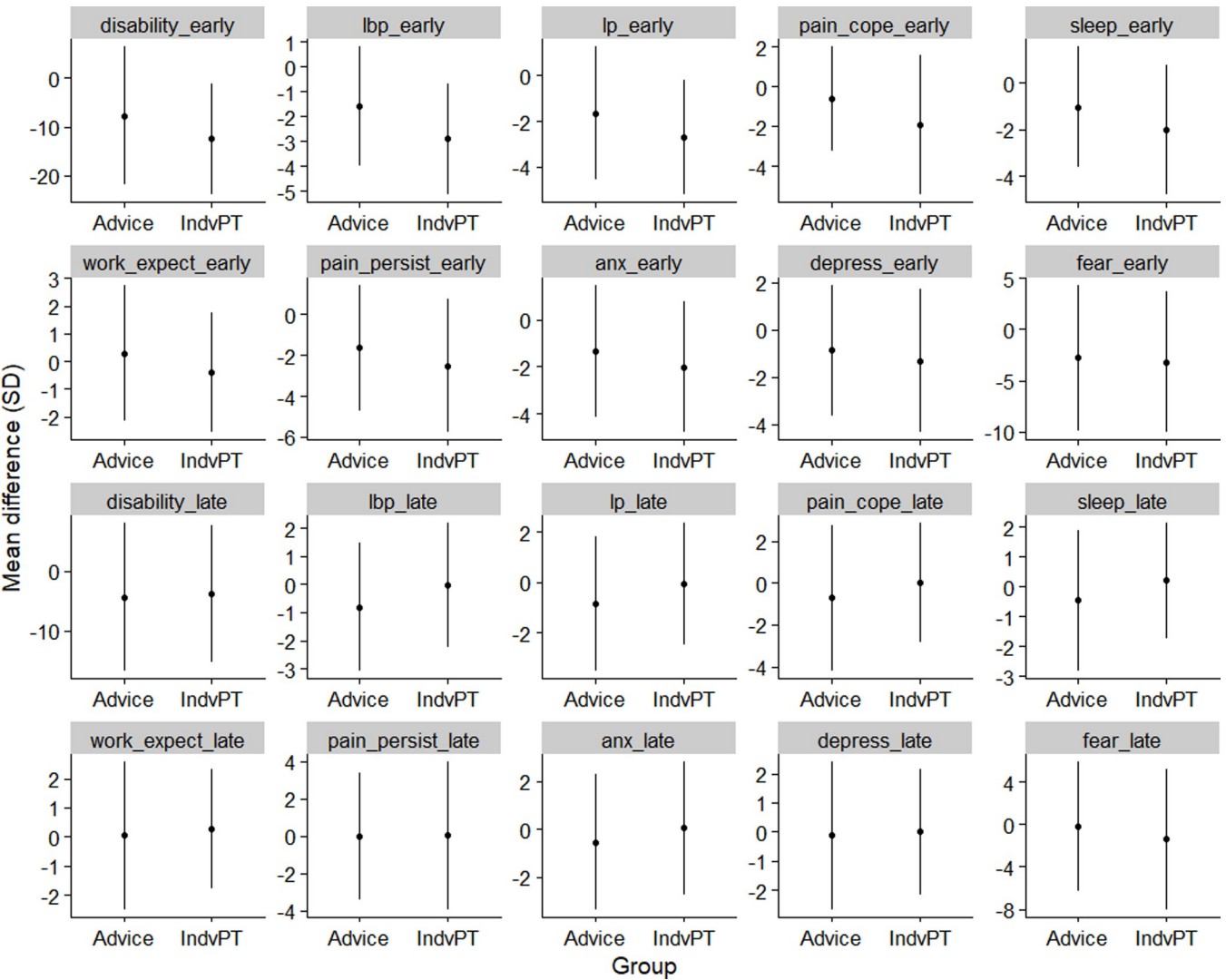

**Fig 1. Mean with error bars (one standard deviation) of the change values of each modelled variable in the Bayesian Network.** Abbreviation: Suffix with "_early"–change score between baseline and week-10 follow-up; "_late"–change value between week-10 and week-52; Anx—anxiety; lbp–low back pain; lp–leg pain; IndvPT–individualised physiotherapy.

Table 2, which varied from low (late change value of work expectations) to high (late change value of disability).

An advantage of BNs is that the model enables the reader to query different elements of the system to fully understand the interaction between variables. By systematically fixing the individual values of some variables from the model, the impact of those variables on the remaining variables in the model becomes clearer. This is akin to the concept of covariate adjustment in traditional multiple regression analysis. For simplicity, the magnitude of relationship between variables are reported using $\beta$ coefficients, which can be interpreted as a one-unit change in the independent variable resulting in a $\beta$ unit change in the dependent variable. The $\beta$ coefficients can be interpreted similar to the path coefficients used in SEM. As the BN model was complex with multiple interconnected variables, we explore in more detail below three main findings that were of most relevance to understanding the recovery pathways in people with LBP receiving individualised physiotherapy versus advice.

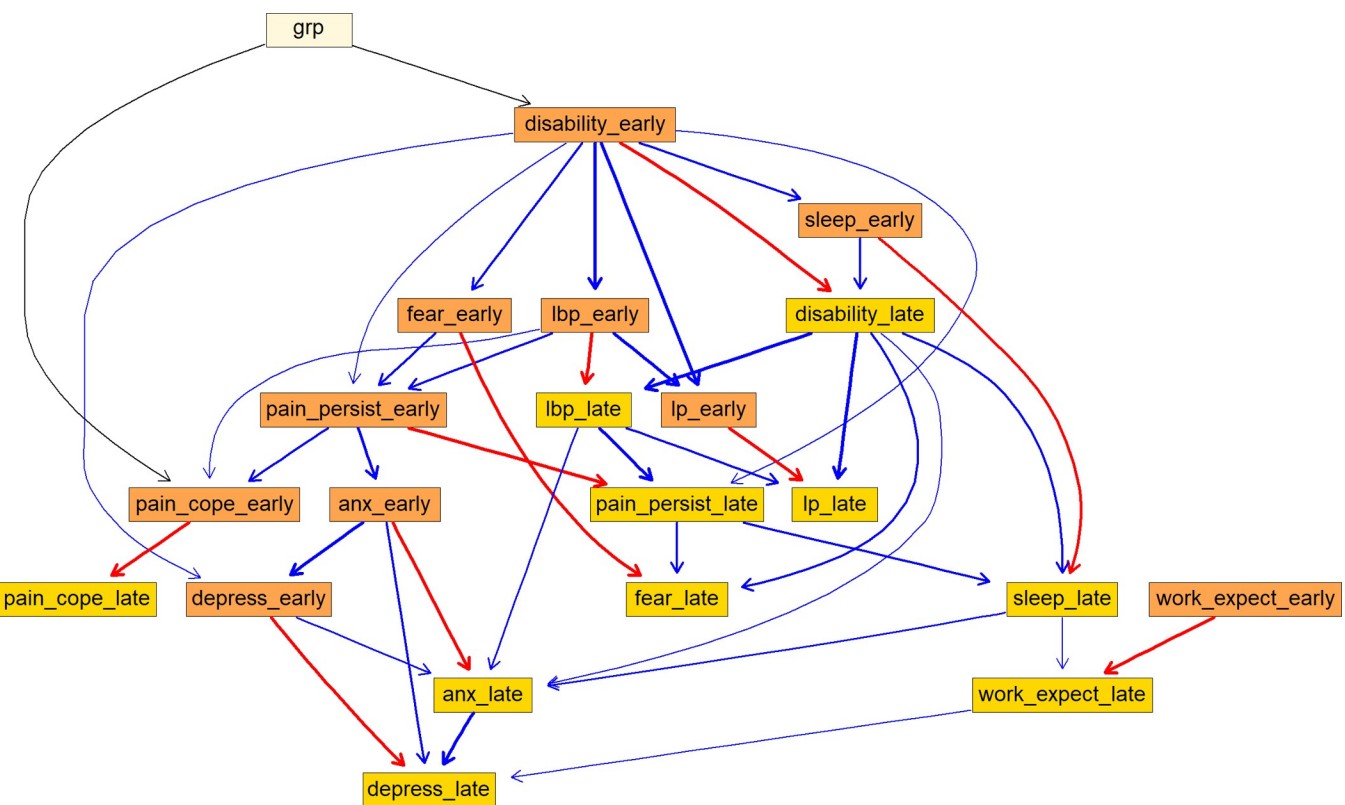

**Fig 2. The directed acyclic graph (DAG) underlying the consensus Bayesian Network learned from the variables across 289 participants.** The colour of the arcs reflect the sign (positive [blue] or negative [red]) of the β coefficient value relating the "parent" to "child" variables. The thickness of the arcs reflects the proportion of times each arc was found in the 200 Bayesian Network models built; only arcs that appeared in >50% of the networks are included in the final averaged consensus network. Abbreviation: Suffix with "_early"–change score between baseline and week-10 follow-up; "_late"–change value between week-10 and week-52; Anx—anxiety; lbp–low back pain; lp–leg pain; IndvPT–individualised physiotherapy.

## Individualised physiotherapy worked predominately by facilitating early changes in disability

The BN showed that individualised physiotherapy exerted direct treatment effects on early change in disability and early change in pain coping (Fig 2). Individualised physiotherapy also indirectly improved back pain intensity, recovery expectations, sleep, fear, anxiety and depression *via* its influence on early change in disability. For example, individualised physiotherapy resulted in a 0.42 unit greater early reduction in LBP intensity than advice (t = -8.68, P < 0.001), but this relationship was dependent on early change in disability. When we simulated a scenario where the treatment group-to-early disability pathway was removed by fixing the value of the early disability regression coefficient to zero, there was no remaining difference between the treatment groups in early reduction in LBP (t = 1.15, P = 0.249). The influence of individualised physiotherapy on early change in LBP was therefore dependent on an early change in disability.

## Pathways from early change in disability to early change in depression

A one-unit early reduction in disability resulted in a $\beta_{disability_{early}}^{depress_{early}}$ = 0.045 unit early reduction in depression (t = 20.38, P < 0.001) (Fig 3). From Fig 2, it can be observed that this relationship was dependent on three variables: early changes in fear, LBP intensity, and pain persistence. When the value of each of the three aforementioned regression coefficients were sequentially

**Table 2. Measurement errors between observed and predicted change values.**

| Variable | Correlation | Correlation strength | RMSE | MSE | MAE |
|---|---|---|---|---|---|
| disability_early | 0.72 | high | 9.2 | 85.4 | 7.2 |
| lbp_early | 0.68 | moderate | 1.8 | 3.4 | 1.4 |
| lp_early | 0.71 | high | 1.9 | 3.5 | 1.4 |
| pain_cope_early | 0.52 | moderate | 2.6 | 7.2 | 2.1 |
| sleep_early | 0.53 | moderate | 1.9 | 3.8 | 1.5 |
| work_expect_early | 0.44 | low | 2.1 | 4.7 | 1.6 |
| pain_persist_early | 0.62 | moderate | 2.4 | 5.9 | 1.9 |
| anx_early | 0.76 | high | 2.0 | 4.4 | 1.2 |
| depress_early | 0.69 | moderate | 2.3 | 5.4 | 1.8 |
| fear_early | 0.55 | moderate | 5.7 | 32.1 | 4.3 |
| disability_late | 0.73 | high | 8.2 | 68.8 | 6.2 |
| lbp_late | 0.71 | high | 1.6 | 2.5 | 1.2 |
| lp_late | 0.70 | moderate | 1.7 | 3.1 | 1.2 |
| pain_cope_late | 0.46 | low | 2.6 | 6.8 | 1.8 |
| sleep_late | 0.59 | moderate | 1.7 | 3.1 | 1.2 |
| work_expect_late | 0.42 | low | 1.6 | 2.7 | 1.1 |
| pain_persist_late | 0.65 | moderate | 2.6 | 7.0 | 1.9 |
| anx_late | 0.74 | high | 1.9 | 4.0 | 1.0 |
| depress_late | 0.65 | moderate | 1.7 | 2.9 | 1.1 |
| fear_late | 0.53 | moderate | 5.0 | 25.4 | 3.8 |

**Abbreviation:** Suffix with "_early"–change score between baseline and week-10 follow-up; "_late"–change value between week-10 and week-52; Anx—anxiety; lbp–low back pain; lp–leg pain; RMSE–root-mean-squared error; MSE- mean squared error; MAE- mean absolute error.

fixed to zero, the $\beta^{depress_{early}}_{disability_{early}}$ coefficient changed to 0.042 (when fixing fear_early, t = 19.23, P < 0.001), 0.038 (when fixing LBP_early, t = 17.70, P < 0.001), and 0.028 (when fixing pain persistence_early, t = 12.88, P = 0.395) (Fig 4). The results of these simulations suggest that

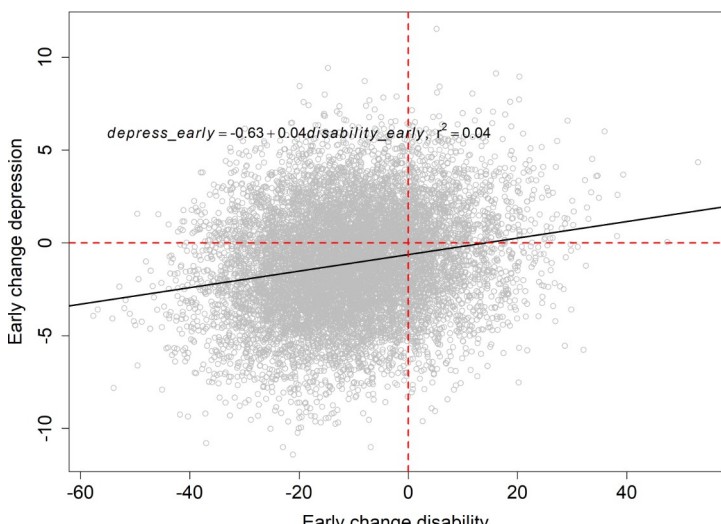

**Fig 3. Posterior samples (104) of the relationship between an early change in disability against an early change in depressive symptoms, with associated linear relationship.**

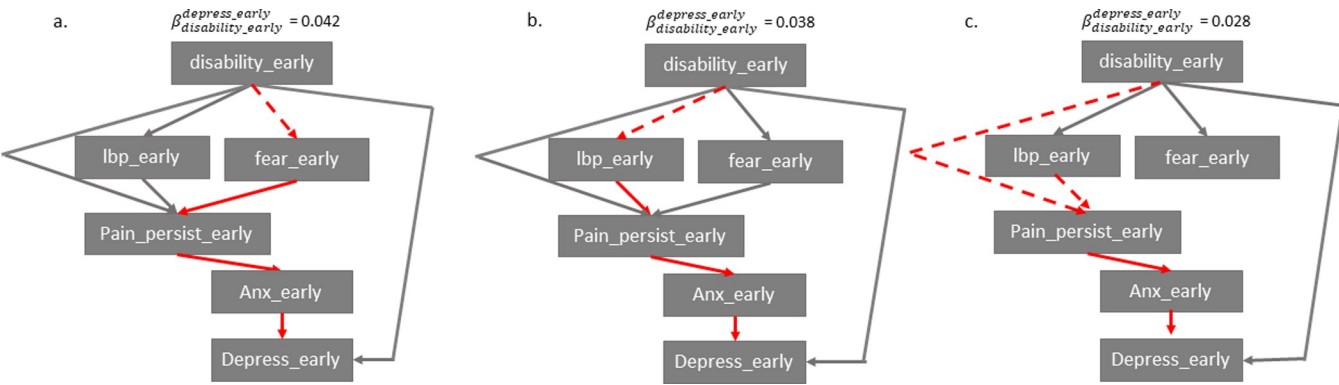

**Fig 4. Contribution of different pathways from early change in disability to early change in depressive symptoms.** A dotted arc (-—>) reflects a simulation whereby the "child" variable was made independent of its "parent", meaning it is removed. Arrows in red reflect the arcs which are "removed" as a consequence of the aforementioned simulation. Information can only pass through grey arcs (not red). (a) scenario where early change in fear was independent of early change in disability, (b) scenario where early change in persistence was independent of early change in disability and low back pain. Abbreviation: Suffix with "_early"–change score between baseline and week-10 follow-up; "_late"–change value between week-10 and week-52; Anx—anxiety; lbp–low back pain.

approximately 7% of the $\beta_{disability_{early}}^{depress_{early}}$ effect flowed through the pathway from fear-to-pain persistence-to-anxiety (Fig 4A), approximately 17% of the effect flowed through the pathway from LBP-to-pain persistence-to-anxiety (Fig 4B), and approximately 38% of the effect flowed through the pathway from pain persistence-to-anxiety (Fig 4C). The remaining 38% of the $\beta_{disability_{early}}^{depress_{early}}$ effect was attributable to the direct effect of disability on depression.

## Relationship between early and late change scores of each variable

It can be observed from Fig 2 that there is a negative relationship between the early and late change scores of the same variable. For example, a one-unit early reduction in LBP was associated with 0.43 *lesser* unit reduction in late LBP change score (t = -51.33, P < 0.001) (Fig 5). That is, participants who experienced substantial early reduction in LBP during treatment

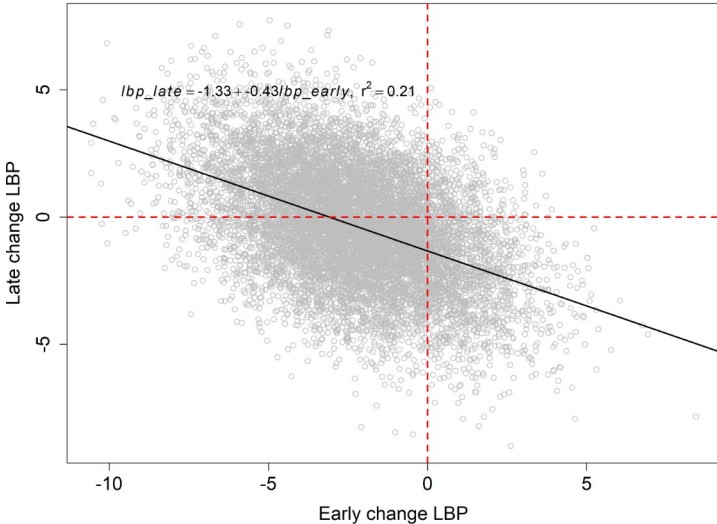

**Fig 5. Posterior samples (104) of the relationship between an early change in disability against an early change in depressive symptoms, with associated linear relationship.**

(between baseline and 10-week follow-up) tended to achieve *smaller additional* improvement in their LBP after discharge (between 10-weeks and 52-weeks).

## Discussion

This is the first study to adopt a data-driven modelling approach to evaluate how individualised physiotherapy works (relative to advice) to help people with LBP. We found that individualised physiotherapy directly reduced disability and pain coping during the intervention period, with early change in disability subsequently facilitating changes in LBP intensity, recovery expectations, sleep, fear, anxiety and depression. We found that early improvement in disability led to early reduction in depression both directly and via more complex pathways involving fear, recovery expectations, anxiety, and LBP intensity. Our findings suggest that individualised physiotherapy treatment effects involve complex interactions between multiple variables that are mainly influential during the intervention period.

The finding that changes in pain and psychosocial variables such as fear, anxiety and depression were dependent on an early change in disability is in contrast to previous studies exploring individualised or stratified physiotherapy [17, 18], as well as the wider literature [45]. Mediation studies to date in the LBP field have tended to use hypothesis-driven modelling techniques which require a model structure to be imposed on the data. As disability is a common primary outcome measure in LBP clinical trials, it has typically been modelled as an "outcome" in mediation studies, hence being dependent on changes in psychosocial variables [17, 18]. It is not implausible that improvement in disability could lead to improvement in psychosocial variables, and that was the finding that emerged from our BN model which did not impose a fixed structure on variables. Other research has pointed to the potential for a reciprocal relationship between depression and pain severity in LBP [46], suggesting that the direction of association between variables may not be straightforward.

Apart from two early change variables (disability and anxiety), the relationships uncovered between early and late change indices were predominantly auto-correlations (i.e. early change was inversely associated with late change of the same variable) (Fig 2). Participants with large early improvements in variables between baseline and 10 weeks (the end of treatment) tended to achieve smaller *additional* improvements following discharge between 10–52 weeks. This finding could have two explanations. It could be that having achieved substantial recovery in the first 10 weeks there was little remaining scope for further improvement in fast-responding participants. Participants who received individualised physiotherapy experienced the greatest benefit within the 10 week treatment period (S2 Fig). It could also suggest that participants less responsive to treatment in the early stages achieved further gradual recovery after discharge– evidenced more clearly in the advice group (S2 Fig). Given that there were no direct influences of group on late change variables, it is likely that the later improvements in the advice group could be attributed to natural recovery (i.e. auto-correlations). These findings could alternatively suggest that strategies to maintain progress and motivation in participants after significant early improvement could be beneficial–such as booster sessions after discharge or preparing participants for long-term independent self-management to ensure ongoing progress [47].

The sequential pathway of the fear-avoidance model (FAM) formulated by Vlaeyen et al. [19] would predict that a change in fear would drive a later change in anxiety, and an even later change in depression. In the present study, only one relationship from the FAM (early anxiety-to-late depression) was found. It is possible that more relationships between the change values of different variables at different times could be due to the assessment intervals being too long [48]. For example, it is quite reasonable to expect that fluctuations in fear alters

anxiety levels within a short timeframe (e.g. hours or even minutes). This means that the relatively long assessment intervals in the current study could largely contain auto-correlations, rather than temporal relations between variables [49]. Future studies may benefit from more frequent data collection strategies, such as daily text messages [50], to more accurately account for the temporal dependency of the modelled variables.

This study had several strengths. First, the data for the BN analysis was obtained from a robustly conducted, large RCT, with 96% of the original cohort included in the present analysis. This is in contrast to previous mediation studies with either a smaller sample size [40], or where the number of drop-outs was high (e.g. up to 37% [17]). Secondly, we report the full codes used in the present analysis, which provides reporting transparency. Third, we discovered some novel recovery pathways that have not been previously investigated in mediation studies, such as pain being dependent on disability and anxiety being dependent on pain intensity.

Despite its strengths, this study is not without limitations. We first note that methods to quantify pathways of change (i.e. causal inference) is not a singular method of scientific query, but actually comprise three hierarchical levels [51]–association (i.e. "how are the variables related?"), intervention (i.e. "what would Y be if I do X?"), and lastly counterfactuals (i.e. "What if I had acted differently?"). Each ladder provides an incremental amount of evidence towards causal inference. Many hypothesis-driven causal mediation analysis methods, as used in contemporary mediation studies [17, 44], operate at the level of counterfactual analysis. The current BN analysis can only act on the first and second rung (via our simulated intervention analysis) of the causation ladder, meaning that we cannot definitively conclude our uncovered pathways are causal. Despite an inability to perform counterfactual analysis, the present analysis could be said to provide competing, and potentially more probable, pathways of effect than those presented in the mediation literature, but this requires future confirmatory research. Another limitation of the present study was that BN model performance ranged from low to high. The lower performance for some variables such as work expectation and pain coping could indicate that there are potentially important factors that influence these variables missing from the current analysis. The STOPS trial prognostic study [52] showed that a range of biological factors were important predictors of outcome (eg. inflammatory indicators, motor control patterns, and range-of-motion measures), but these variables were only measured at baseline preventing the evaluation of their role as potentially important mechanisms operating between measured variables in the current study. For example, our model cannot exclude the possibility that individualised physiotherapy improved disability *via* latent mechanisms that were not measured in the current study (such as tissue healing or reduced inflammation). Including a wider range of biopsychosocial factors into the BN analysis may improve the overall network model's correlation performance.

## Conclusions

A data-driven BN modelling approach showed that individualised physiotherapy directly reduced disability and pain coping during the intervention period in people with low back pain. The effects of individualised physiotherapy on pain intensity, recovery expectations, sleep, fear, anxiety and depression were all exerted *via* an early change in disability. Early improvement in disability led to early reduction in depression both directly and via more complex pathways involving fear, recovery expectations, anxiety and LBP intensity. The effects of individualised physiotherapy on the investigated variables were most influential during the intervention period (0–10 weeks). Our findings suggest that the pathways to recovery from LBP in people receiving individualised physiotherapy are complex and multifactorial. Future RCTs could consider measuring a broad range of potential biopsychosocialvariables to

facilitate BN analysis which is well suited to exploring complex inter-relationships between multiple variables.

## Supporting information

**S1 Fig. Percentage (out of n = 289 participants) missing data for each of the unfolded variables included in the Bayesian Network.** Abbreviation: Suffix with "_early"–change score between baseline and week-10 follow-up; "_late"–change value between week-10 and week-52; Anx—anxiety; grp–group; lbp–low back pain; lp–leg pain.
(TIFF)

**S2 Fig. Mean with error bars as one standard deviation of the original values of each modelled variables in the Bayesian Network at each follow up time point.** Abbreviation: Anx—anxiety; lbp–low back pain; lp–leg pain; IndvPT–individualized physiotherapy.
(TIFF)

## Acknowledgments

The authors wish to acknowledge the trial physiotherapists who volunteered to treat participants in this trial free of charge.

## Author Contributions

**Conceptualization:** Bernard X. W. Liew, Andrew J. Hahne.

**Data curation:** Jon J. Ford.

**Formal analysis:** Bernard X. W. Liew, Marco Scutari, Andrew J. Hahne.

**Funding acquisition:** Jon J. Ford, Andrew J. Hahne.

**Investigation:** Jon J. Ford, Andrew J. Hahne.

**Methodology:** Bernard X. W. Liew, Jon J. Ford, Marco Scutari, Andrew J. Hahne.

**Project administration:** Jon J. Ford, Andrew J. Hahne.

**Software:** Marco Scutari.

**Supervision:** Jon J. Ford, Andrew J. Hahne.

**Validation:** Bernard X. W. Liew, Marco Scutari.

**Visualization:** Bernard X. W. Liew.

**Writing – original draft:** Bernard X. W. Liew, Jon J. Ford, Marco Scutari, Andrew J. Hahne.

**Writing – review & editing:** Bernard X. W. Liew, Jon J. Ford, Marco Scutari, Andrew J. Hahne.

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
