## [Decision Letter · Decision Letter 0]

26 Aug 2021

PONE-D-21-22271

Using data-driven Bayesian Network analysis to explore recovery pathways in people with low back pain receiving individualised physiotherapy or advice

PLOS ONE

Dear Dr. Liew,

Thank you for submitting your manuscript to PLOS ONE. After careful consideration, we feel that it has merit but does not fully meet PLOS ONE’s publication criteria as it currently stands. Therefore, we invite you to submit a revised version of the manuscript that addresses the points raised during the review process.

We look forward to receiving your revised manuscript.

Kind regards,

Walid Kamal Abdelbasset, Ph.D.

Academic Editor

PLOS ONE

Journal Requirements:

2. Please provide additional details regarding participant consent. In the Methods section, please ensure that you have specified (1) whether consent was informed and (2) what type you obtained (for instance, written or verbal). If your study included minors, state whether you obtained consent from parents or guardians. If the need for consent was waived by the ethics committee, please include this information.

Reviewers' comments:

Reviewer's Responses to Questions

**Comments to the Author**

1. Is the manuscript technically sound, and do the data support the conclusions?

Reviewer #1: Yes

Reviewer #2: Yes

2. Has the statistical analysis been performed appropriately and rigorously? 

Reviewer #1: No

Reviewer #2: Yes

3. Have the authors made all data underlying the findings in their manuscript fully available?

Reviewer #1: No

Reviewer #2: No

4. Is the manuscript presented in an intelligible fashion and written in standard English?

Reviewer #1: No

Reviewer #2: Yes

5. Review Comments to the Author

Reviewer #1: This manuscript requires a significant amount of improvement in:

Title

1. Title needs to be modified.

2. Kindly frame title such that it is accurate, informative, descriptive, succinct, simple and specific.

3. Kindly avoid the words like

Abstract:

1. Trial design is not mentioned.

2. The background needs to be shortened.

3. Methods section is poorly framed. It has to be re-written.

4. Demographic profile of patients is not mentioned.

Introduction: please follow these steps to have scientific introduction

1. Explain the rationale of the study. Please delete information unrelated to objective so that the section is short and sweet. For example, the first page of introduction may be deleted. Kindly focus on three elements of introduction.

a. What is known about the topic? (Background)

b. What is not known? (The research problem)

c. Why the study was done? (Justification)

2. Objective is not clear as mentioned above.

1. Methods section determines the results. Kindly focus on three basic elements of methods section.

a. How the study was designed?

b. How the study was carried out?

c. How the data were analyzed?

d. Components of methods

i. Study design, setting, sample size

ii. Participant

iii. Intervention/issue of interest (exposure)

iv. Comparison

v. Ethics and end point

vi. Statistical analysis

1. The discussion section needs to be described scientifically. Kindly frame it along the following lines:

i. Main findings of the present study

ii. Comparison with other studies

iii. Implication and explanation of findings

iv. Strengths and limitations need to be clear

v. Conclusion, recommendation and future direction.

Reviewer #2: The authors conduct a Bayesian Network approach to model the recovery pathways in 300 people with low back pain receiving individualized physiotherapy or advice from a randomized controlled trial. The results showed that individualized physiotherapy reduced early improvement in disability and its relevant consequences.

1. Variables included in the Bayesian Network. Was sample characteristic information (e.g. age, sex, and so on) included, either as nodes or adjusted, in the Network construction as these information may play an important role?

2. Please comment on whether the missingness behave differently for the participants in two different groups.

3. Table 2. Correlation was reported. What measure of correlation was reported? If Pearson correlation was reported, were the data normally distributed?

4. Please comment on how the directions were decided. Any potential misspecification or reciprocal relationship? If so, how these may affect the results?

6. PLOS authors have the option to publish the peer review history of their article (what does this mean?). If published, this will include your full peer review and any attached files.

Reviewer #1: No

Reviewer #2: No

---

## [Author Response · Author response to Decision Letter 0]

7 Sep 2021

Please see the uploaded Response to Reviewers document for a proper view of our detailed reply, which includes tables, figures with the appropriate formatting.

Editor

Thank-you for the opportunity to revise our manuscript for PLOS ONE. We have carefully addressed the comments of the editor and the two reviewers, and feel confident that the manuscript has been enhanced by these recommendations. 

Reply: We have checked the manuscript against the PLOS ONE style requirements and believe it is compliant. 

2. Please provide additional details regarding participant consent. In the Methods section, please ensure that you have specified (1) whether consent was informed and (2) what type you obtained (for instance, written or verbal). 

Reply: Further information around consent of participants has been added to the manuscript as per the text below (L97): 

All participants provided written informed consent to participate in the trial.

Reply: Participants from the study provided informed consent that their data would only be made available to other researchers on specific request to Dr. Ford or Dr. Hahne. We are therefore unable to publicly publish the dataset from this study, but reasonable requests to access the data can be directed either to the authors, or alternatively to: 

 Senior Manager, Ethics, Integrity and Biosafety 

Human Ethics Committee 

 La Trobe University 

Telephone: +61 3 9479 1443

 Email: humanethics@latrobe.edu.au

 Quote ethics ID: FHEC08/196

Once again, than-you for the opportunity to revise our paper for PLOS ONE. 

Reviewer #1: 

This manuscript requires a significant amount of improvement in:

Title

1. Title needs to be modified.

Reply: The title has been completely re-written

Changed from: Using data-driven Bayesian Network analysis to explore recovery pathways in people with low back pain receiving individualised physiotherapy or advice.

Changed to: How does individualised physiotherapy work for people with low back pain? A Bayesian Network analysis using randomised controlled trial data. 

2. Kindly frame title such that it is accurate, informative, descriptive, succinct, simple and specific.

Reply: We thank the Reviewer for this specific suggestion. The new title as presented above has attempted to address these criteria. 

3. Kindly avoid the words like

Reply: The comment was incomplete, but as the title has been completely re-written we hope the concern has been addressed. 

Abstract:

1. Trial design is not mentioned.

Reply: We thank the Reviewer for this observation. The trial design has been added to the abstract using the following text (L22): 

We sought to determine how this treatment works by using randomised controlled trial data to develop a Bayesian Network model. 

2. The background needs to be shortened.

Reply: We thank the Reviewer for their suggestion. We have shortened the background in the Abstract to comprise two short sentences as indicated below (L21): 

Individualised physiotherapy is an effective treatment for low back pain. We sought to determine how this treatment works by using randomised controlled trial data to develop a Bayesian Network model. 

3. Methods section is poorly framed. It has to be re-written.

Reply: We have completely re-structured the methods section of the Abstract to make it clearer, incorporating information on participants, treatments, analysis approach, and what the data is able to explain (L24). 

300 randomised controlled trial participants (153 male, 147 female, mean age 44.1) with low back pain (of duration 6-26 weeks) received either individualised physiotherapy or advice. Variables with potential to explain how individualised physiotherapy works were included in a multivariate Bayesian Network model. Modelling incorporated the intervention period (0-10 weeks after study commencement – “early” changes) and the followup period (10-52 weeks after study commencement – “late” changes). Sequences of variables in the Bayesian Network showed the most common direct and indirect recovery pathways followed by participants with low back pain receiving individualised physiotherapy or advice.

4. Demographic profile of patients is not mentioned.

Reply: This has been added to the methods section of the Abstract (L24):

300 randomised controlled trial participants (153 male, 147 female, mean age 44.1) with low back pain (of duration 6-26 weeks)…

Introduction: please follow these steps to have scientific introduction

1. Explain the rationale of the study. Please delete information unrelated to objective so that the section is short and sweet. For example, the first page of introduction may be deleted. Kindly focus on three elements of introduction.

a. What is known about the topic? (Background)

b. What is not known? (The research problem)

c. Why the study was done? (Justification)

Reply: We thank the Reviewer for their helpful suggestions to help focus the structure of our Introduction. We have re-written the Introduction to focus on 1) introducing the topic (how do low back pain treatments work), 2) presenting the limitations of previous research that has tried to answer this question, and establishing what questions remain unanswered, and 3) justification and objectives of the current study. 

2. Objective is not clear as mentioned above.

Reply: As part of re-writing the Introduction, the objective of the study is now more clearly stated at the end of the introduction (L88). 

The aim of this study was to determine how the individualised physiotherapy (relative to the advice) approach in the STOPS RCT helped people with LBP. 

1. Methods section determines the results. Kindly focus on three basic elements of methods section.

a. How the study was designed?

b. How the study was carried out?

c. How the data were analyzed?

d. Components of methods

i. Study design, setting, sample size

ii. Participant

iii. Intervention/issue of interest (exposure)

iv. Comparison

v. Ethics and end point

vi. Statistical analysis

Reply: We thank the Reviewer for their recommendations for important elements to include in the Methods section. We have revised the manuscript to ensure that each suggested element is included. We have also checked recent articles in PLOS ONE to ensure consistency of structure and subheadings. Changes to the manuscript as a result of this restructuring are evident via tracked changes. 

1. The discussion section needs to be described scientifically. Kindly frame it along the following lines:

i. Main findings of the present study

ii. Comparison with other studies

iii. Implication and explanation of findings

iv. Strengths and limitations need to be clear

v. Conclusion, recommendation, and future direction.

Reply: We thank the Reviewer again for outlining the important elements for Discussion. We have revised our manuscript to ensure that all suggested elements are addressed. However, given that we discuss three main points a slightly altered structure was chosen to the one suggested by the Reviewer so that each main point could be thoroughly discussed one at a time. The new structure involves: 

 Paragraph 1 summarises the main findings of the study from the results 

Paragraph 2 discusses the first main finding in greater depth, including a comparison to other studies and explanation/implications of findings

Paragraph 3 discusses the second main finding in greater depth, including a comparison to other studies and explanation/implications of findings

Paragraph 4 discusses the third main finding in greater depth, including a comparison to other studies and explanation/implications of findings

Paragraph 5 discusses the strengths of the study 

Paragraph 6 discusses the limitations of the study 

Paragraph 7 concludes and makes recommendations for future research

We believe this structure flows logically, aligns with the order of the results presented, and is consistent with the structure of other recent articles published in PLOS ONE. 

Reviewer #2

The authors conduct a Bayesian Network approach to model the recovery pathways in 300 people with low back pain receiving individualized physiotherapy or advice from a randomized controlled trial. The results showed that individualized physiotherapy reduced early improvement in disability and its relevant consequences.

1. Variables included in the Bayesian Network. Was sample characteristic information (e.g. age, sex, and so on) included, either as nodes or adjusted, in the Network construction as these information may play an important role?

Reply: We only included factors in the Bayesian Network that were potentially modifiable, as this would make them potentially useful therapeutic targets. Sample characteristics at baseline are not modifiable with treatment. In addition, our previous publications on the current data set show that there were no differences between treatment groups on any of the baseline sample characteristic (Ford et al. 2016), nor were these characteristics predictive of outcome (Ford et al 2018). However, we have also included the descriptive statistics of the baseline demographic characteristics in the revised Table 1, in response to the comment below. 

We also ran an exploratory Bayesian Network analysis with the four demographic variables included, but found that these variables did not influence other variables (see image below). To create a simpler model, we excluded these four demographic variables during the formal analysis.

2. Please comment on whether the missingness behave differently for the participants in two different groups.

Reply: We thank the Reviewer for this comment. We have included in the revised Table 1, baseline descriptive characteristics for all participants, including the 11 that were excluded due to high missing data.

Table 1.Baseline descriptive characteristics of cohort included in Bayesian Network analysis

Baseline variable Advice (n = 137) Individualized PT (n = 152) Missing-excluded (n = 11) P.value

Age(years) 44.69(12.33) 42.67(12.02) 44.19(11.75) 0.368

Sex-Female* 67 74 6 0.931

Sex-Male* 70 78 5 0.931

Weight(kg) 77.41(15.79) 77.26(17.77) 174.38(6.07) 0.195

Height(cm) 169.97(11.52) 172.24(10.47) 74.7(8.93) 0.885

Disability 29.71(13.25) 29.31(11.85) 26.82(5.78) 0.750

Low back pain intensity 5.48(1.94) 5.29(1.97) 4.77(1.47) 0.415

Leg pain intensity 4.76(2.62) 4.58(2.74) 4.8(2.49) 0.860

Pain coping 5.06(2.15) 5.16(2.37) 4.36(2.38) 0.526

Anxiety 5.08(2.5) 4.62(2.58) 4(3.52) 0.183

Depression 3.54(2.98) 3.01(2.98) 3.64(3.5) 0.312

Pain persistence 7.26(2.17) 6.74(2.36) 6.55(3.17) 0.128

Work expectations 2.01(3.24) 1.63(2.95) 5.36(4.18) 0.001

Sleep 3.69(2.54) 3.57(2.49) 3.45(2.5) 0.893

Fear 18.11(6.01) 18.58(6.17) 22.6(6.13) 0.081

Abbreviations: PT - physiotherapy

* - variables represent counts of participants. All other values reflect mean (one standard deviation).

P values for continuous variables derived from a one-way Analysis of Variance test, count variable from a Chi-squared Test.

3. Table 2. Correlation was reported. What measure of correlation was reported? If Pearson correlation was reported, were the data normally distributed?

Reply: We thank the Reviewer for this comment. Yes, this was the Pearson correlation, and we edited the wording to reflect this change. The assumption of the normal distribution is really important only if the statistical inference of the correlation was undertaken – which we did not in the present project. Furthermore, we confirmed that the reported Pearson correlations are not inflated: both the observed and the predicted values are approximately symmetric and present little in the way of skewness. We also excluded inflation due to outliers by checking a plot of observed values against predicted values. In addition to the correlation values, we have included other traditional metrics of performance in linear regression modelling, such as root-mean-square error, mean square error, and mean absolute error (revised Table 2).

Table 2 Measurement errors between observed and predicted change values

Variable Correlation Correlation strength RMSE MSE MAE

disability_early 0.72 high 9.2 85.4 7.2

lbp_early 0.68 moderate 1.8 3.4 1.4

lp_early 0.71 high 1.9 3.5 1.4

pain_cope_early 0.52 moderate 2.6 7.2 2.1

sleep_early 0.53 moderate 1.9 3.8 1.5

work_expect_early 0.44 low 2.1 4.7 1.6

pain_persist_early 0.62 moderate 2.4 5.9 1.9

anx_early 0.76 high 2.0 4.4 1.2

depress_early 0.69 moderate 2.3 5.4 1.8

fear_early 0.55 moderate 5.7 32.1 4.3

disability_late 0.73 high 8.2 68.8 6.2

lbp_late 0.71 high 1.6 2.5 1.2

lp_late 0.70 moderate 1.7 3.1 1.2

pain_cope_late 0.46 low 2.6 6.8 1.8

sleep_late 0.59 moderate 1.7 3.1 1.2

work_expect_late 0.42 low 1.6 2.7 1.1

pain_persist_late 0.65 moderate 2.6 7.0 1.9

anx_late 0.74 high 1.9 4.0 1.0

depress_late 0.65 moderate 1.7 2.9 1.1

fear_late 0.53 moderate 5.0 25.4 3.8

Abbreviation: Abbreviation: Suffix with “_early” – change score between baseline and week-10 follow-up; “_late” – change value between week-10 and week-52; Anx - anxiety; lbp – low back pain; lp – leg pain; RMSE – root-mean-squared error; MSE- mean squared error; MAE- mean absolute error.

4. Please comment on how the directions were decided. Any potential misspecification or reciprocal relationship? If so, how these may affect the results?

Reply: We thank the Reviewer for this comment. Herein, we use the term “structural learning” to reflect learning the direction of relationships. We used the “hill-climbing” algorithm, which was embedded into structural expectation-maximization for structural learning. The hill-climbing algorithm uses a search strategy that explores the space of the directed acyclic graphs by single-arc addition, removal, and reversals (without introducing cycles into the graph); with random restarts to avoid local optima. The structure that maximizes the model fit to the data is retained. 

Reciprocal, bidirectional, relationships are not permitted in a Bayesian Network model, because by definition, a Bayesian Network models an acyclic graph. Bidirectional relationships will introduce cycles into the model, something that is avoided in all structural learning algorithms for Bayesian Network. Note that cyclic and bidirectional/reciprocal relationships are forbidden in the Bayesian networks, in which the same variable is represented by different nodes at different time points; but they can still be read from the network if we consider variables after abstracting away time. For instance, in Figure 2 we can see the arcs anx_early -> depress_late and depress_early -> anx_late: these indicate a feedback loop in which anxiety feeds depression which in turn feeds anxiety.

Misspecification is possible if there are missing confounding variables. We have identified this as a potential study limitation in the Discussion of the Original manuscript. To further mitigate modelling biologically unrealistic relationships (e.g. if a variable at 12th month can influence another variable at baseline), we included a blacklist into the model, so that such relationships are enforced during model building.

---

## [Decision Letter · Decision Letter 1]

29 Sep 2021

How does individualised physiotherapy work for people with low back pain? A Bayesian Network analysis using randomised controlled trial data

PONE-D-21-22271R1

Dear Dr. Liew,

We’re pleased to inform you that your manuscript has been judged scientifically suitable for publication and will be formally accepted for publication once it meets all outstanding technical requirements.

Kind regards,

Walid Kamal Abdelbasset, Ph.D.

Academic Editor

PLOS ONE

Additional Editor Comments (optional):

Reviewers' comments:

Reviewer's Responses to Questions

**Comments to the Author**

1. If the authors have adequately addressed your comments raised in a previous round of review and you feel that this manuscript is now acceptable for publication, you may indicate that here to bypass the “Comments to the Author” section, enter your conflict of interest statement in the “Confidential to Editor” section, and submit your "Accept" recommendation.

Reviewer #1: All comments have been addressed

Reviewer #2: All comments have been addressed

2. Is the manuscript technically sound, and do the data support the conclusions?

Reviewer #1: Yes

Reviewer #2: (No Response)

3. Has the statistical analysis been performed appropriately and rigorously? 

Reviewer #1: Yes

Reviewer #2: (No Response)

4. Have the authors made all data underlying the findings in their manuscript fully available?

Reviewer #1: Yes

Reviewer #2: (No Response)

5. Is the manuscript presented in an intelligible fashion and written in standard English?

Reviewer #1: Yes

Reviewer #2: (No Response)

6. Review Comments to the Author

Reviewer #1: (No Response)

Reviewer #2: (No Response)

7. PLOS authors have the option to publish the peer review history of their article (what does this mean?). If published, this will include your full peer review and any attached files.

Reviewer #1: No

Reviewer #2: No

---

## [Editor Report · Acceptance letter]

1 Oct 2021

PONE-D-21-22271R1 

How does individualised physiotherapy work for people with low back pain? A Bayesian Network analysis using randomised controlled trial data.   

Dear Dr. Liew:

I'm pleased to inform you that your manuscript has been deemed suitable for publication in PLOS ONE. Congratulations! Your manuscript is now with our production department. 

Kind regards, 

on behalf of

Dr. Walid Kamal Abdelbasset 

Academic Editor

PLOS ONE